# Emergent quantum phase transition of a Josephson junction coupled to a high-impedance multimode resonator

Luca Giacomelli ●[1] & Cristiano Ciuti ●[1] ✉

The physics of a single Josephson junction coupled to a resistive environment is a long-standing fundamental problem at the center of an intense debate, strongly revived by the advent of superconducting platforms with high-impedance multimode resonators. Here we investigate the emergent criticality of a junction coupled to a multimode resonator when the number of modes is increased. We demonstrate how the multimode environment renormalizes the Josephson and capacitive energies of the junction so that in the thermo-dynamic limit the charging energy dominates when the impedance is larger than the resistance quantum and is negligible otherwise, independently from the bare ratio between the two energy scales and the compact or extended nature of the phase of the junction. Via exact diagonalization, we find that the transition surprisingly stems from a level anticrossing involving not the ground state, but the first excited state, whose energy gap vanishes in the thermodynamic limit. We clarify the nature of the two phases by pointing at a different behavior of the ground and excited states and we show that at the transition point the spectrum displays universality not only at low frequencies. In agreement with recent experiments, we reveal striking spectral signatures of the phase transition.

The physics of a quantum system coupled to a bath with many degrees of freedom is a fundamental problem of many-body physics. One of the most famous examples is the spin-boson model for a two-level system coupled to a bath of harmonic oscillators[1,2], exhibiting a localization-delocalization phase transition. A related challenging problem is the resistively shunted Josephson junction[3]. Schmid[4] and Bulgadaev[5] first addressed this problem by considering the junction as a fictitious particle in a periodic cosine potential and the resistor as an infinite bath of harmonic oscillators. The junction was predicted to display a superconducting behavior (phase localization) for shunting resistances $R < R_q$ and an insulating behavior (phase delocalization) for $R > R_q$, where $R_q = h/(2e)^2$ is the resistance quantum, $h$ the Planck constant and $e$ the electron charge (this definition of the resistance quantum differs from the von Klitzing constant of quantum Hall systems by a factor 1/2, linked to the double charge of a Cooper pair). This transition was predicted not to depend on the ratio between the

Josephson energy $E_J$ and the charging energy $E_C$. Following works using perturbative renormalization group[6–10], as well as Monte Carlo methods[11–14], further characterized the transition and confirmed its independence on $E_J/E_C$.

Experimentally, a few works investigated the predicted insulating phase[15–18], but were disputed in a recent experimental work[19] reporting the ac linear response of a resistively shunted junction and claiming no insulating behavior. This work revived the interest in this fundamental problem and sparked a lively debate. Yet, new theoretical analyses did not reach a consensus, with claims of absence[19,20], existence[21,22] and non-trivial modification of the predicted phase diagram[23] (although approaches based on the renormalization group are still a matter of debate[24–26]). Four key questions are outstanding: (i) which are the circuit Hamiltonian terms that are irrelevant in the thermodynamic limit? (ii) does a transition actually happen, and how should one probe the so-called insulating phase? (iii) does the compact versus extended

[1]Université Paris Cité, CNRS, Matériaux et Phénomènes Quantiques, Paris, France. ✉e-mail: cristiano.ciuti@u-paris.fr

nature[27,28] of the Josephson junction phase play a role? (iv) Is the transition independent on $E_J/E_C$?

The interest in this problem has also been catalyzed by impressive experimental advances in circuit quantum electrodynamics (QED)[29], particularly in the study of Josephson junctions coupled to high-impedance environments[30–38]. A recent experiment[39] took a different approach: multimode transmission line resonators were used instead of resistors. Importantly, instead of measuring the transport properties of the junction, this experiment focused on the modifications of the linear-response spectral properties. The resonators used in these experiments are finite-size systems, in particular they have a finite free spectral range (mode frequency spacing). In this context, the thermodynamic limit corresponds to the vanishing free spectral range.

In this work, we investigate theoretically the emergence of the transition for a Josephson junction coupled to a transmission-line resonator when the free spectral range (number of modes) goes to zero (infinity), which, to the best of our knowledge, has never been explored. By approaching finite-size systems with a combination of analytical approaches and numerical exact diagonalization, we show that for a homogeneous transmission line, the transition emerges at $R_q/Z = 1$, where $Z$ is the line characteristic impedance, with the main features being present already for a small number of modes. At the transition point the system displays a universality independent of the ratio $E_J/E_C$, that we confirm with exact solutions not only at small $E_J$. Moreover, unlike a wide class of phase transitions[40], we show that the emergent criticality involves a level anticrossing affecting the first excited state and not the ground state. We also show how the occurrence of the phase transition creates distinctive spectral signatures even at high frequencies and how the ground state behaves differently from the excited states.

## Results

### Quantum circuit model

Let us consider the multimode circuit QED system in Fig. 1a, namely a Josephson junction coupled to a lumped-element transmission line. As detailed in the Methods, the quantum Hamiltonian has been derived from the circuit Lagrangian depending on $\mathbf{C}$ and $\boldsymbol{\Gamma}$, respectively, the capacitance and inductance matrices. The system degrees of freedom are described by the vector $\boldsymbol{\varphi}$, containing independent flux variables[41], indicated in Fig. 1a. To get the final form, we have performed a simultaneous diagonalization of $\mathbf{C}$ and $\boldsymbol{\Gamma}$ via the basis change $\boldsymbol{\varphi} = \mathbf{P}\boldsymbol{\phi}$. Through a Legendre transform and identifying as the junction phase

$\varphi = \sum_k P_{0k}\phi_k$, we have obtained the classical Hamiltonian (for an analogous procedure see[42]). Finally, by quantizing the degrees of freedom, we get the quantum Hamiltonian in the charge gauge

$$\hat{\mathcal{H}} = \hat{\mathcal{H}}_J + \sum_{k=1}^{N_m} \hbar\omega_k \hat{a}_k^\dagger \hat{a}_k + i\hat{N}\sum_{k=1}^{N_m} g_k(\hat{a}_k^\dagger - \hat{a}_k), \quad (1)$$

where $\hat{\mathcal{H}}_J = 4E_C\hat{N}^2 - E_J\cos\hat{\varphi}$ and $\hat{a}_k^\dagger$ is the bosonic creation operator for the $k$th mode and $g_k = 2\sqrt{\hbar\omega_k E_C}P_{0k}$ its coupling to the junction. Here we used the following sum rule, due to the normalization identity $\mathbf{P}^T\mathbf{C}\mathbf{P} = I$,

$$P_{00}^2 + \frac{1}{4E_C}\sum_{k=1}^{N_m}\frac{g_k^2}{\hbar\omega_k} = 1. \quad (2)$$

The frequencies and couplings in (1) depend on the capacitive and inductive matrices. Illustrative values are shown in Fig. 1c, d. A lumped-element circuit possesses an upper frequency cutoff, known as the plasma frequency $\omega_p$. Overall, the line has three key parameters: the impedance $Z = \sqrt{L/C}$, the plasma frequency $\omega_p = 2/\sqrt{LC}$, and the free spectral range $\Delta = \frac{\pi}{\sqrt{LC}}\frac{1}{N_m}$, where $N_m$ is the number of modes. The phase transition is expected to emerge in the thermodynamic limit $N_m \to \infty$, that is equivalent to $\Delta \to 0$.

### Renormalized Josephson and charging energies

It is insightful to consider the basis diagonalizing the last two terms of (1), namely $|N, \mathbf{n}\rangle = |N\rangle \otimes_{k=1}^{N_m} |\alpha_{N,k}, n_k\rangle$, where $|\alpha_{N,k}, n_k\rangle = D_k(\alpha_{N,k})|n_k\rangle$, with $D_k(\alpha) = \exp(\alpha\hat{a}_k^\dagger - \alpha^*\hat{a}_k)$ the displacement operator[43] for the $k$th mode with $\alpha_{N,k} = iN\frac{g_k}{\hbar\omega_k}$, and $\mathbf{n}$ is a vector with the number of photons in each mode. The only nontrivial term in (1) is the junction Hamiltonian: the dressing of the junction by the line can be determined by inspecting its matrix elements in such a basis. As detailed in the Methods, the coupling to the line effectively produces a renormalized Josephson energy

$$\widetilde{E}_J = \exp\left(-\frac{1}{2}\sum_{k=1}^{N_m}\frac{g_k^2}{\hbar\omega_k^2}\right)E_J, \quad (3)$$

independent of the number of photons in the modes. The remaining effect is a renormalized capacitive energy

$$\widetilde{E}_C = P_{00}^2 E_C < E_C. \quad (4)$$

$P_{00}$ vanishes in the thermodynamic limit and has been taken to be exactly zero in a recent study of the phase transition[23]. However, the way it approaches zero in the thermodynamic limit is important and cannot be neglected. Indeed, also the renormalized Josephson energy $\widetilde{E}_J$ tends to zero for $\Delta \to 0$.

The properties of the infinite system will hence be determined by the asymptotic behavior of their ratio. We computed such analytical expression by injecting the numerically evaluated values of $g_k$, $\omega_k$ and $P_{00}$ for different values of the impedance $Z$ and decreasing values of $\Delta$. The results are summarized in Fig. 2. One can see that both $\widetilde{E}_C$ and $\widetilde{E}_J$ decrease for all values of $R_q/Z$ while $\Delta$ is decreased (Fig. 2b, c). However, while $\widetilde{E}_C$ monotonically decreases while increasing $R_q/Z$, $\widetilde{E}_J$ first decreases and then increases. The ratio of the two quantities $\widetilde{E}_J/\widetilde{E}_C$ instead displays a striking behavior (Fig. 2d): for small enough $\Delta$ it sharply decreases from 1 to very small values and then grows again. Remarkably, the curves for different values of $\Delta$ all cross for $Z = R_q$, with the curves for larger sizes (smaller $\Delta$) tending to smaller values for $R_q/Z < 1$, and to larger values for $R_q/Z > 1$. The inset Fig. 2e shows that the slope of the curves at $Z = R_q$ grows logarithmically with $\Delta^{-1}$. We can conclude that in the thermodynamic limit

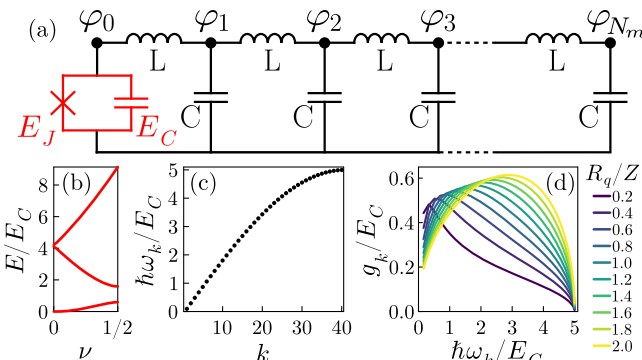

**Fig. 1 | Overview of the system under study. a** Representation of a circuit consisting of a Josephson junction (red) of Josephson energy $E_J$ and capacitive energy $E_C$ coupled to a transmission line resonator. The fluxes used to describe the circuit are indicated with black dots. **b** Energy bands of the uncoupled junction in terms of the quasi-charge $v$ for $E_J = 0.5E_C$. **c** Frequencies of the transmission line modes versus the mode number. **d** Junction-line couplings in the charge gauge as a function of the mode frequency for different values of the impedance $Z$. For these last two panels: plasma frequency $\hbar\omega_p = 5E_C$ and free spectral range $\hbar\Delta = 0.2E_C$ (see definition in the text).

the ratio $\widetilde{E}_J/\widetilde{E}_C$ tends to 0 for $R_q/Z < 1$ (apart from $R_q/Z = 0$, for which no renormalization can occur) and to infinity for $R_q/Z > 1$.

This points at the two expected phases: for $R_q/Z < 1$ the charging energy dominates (so-called insulating behavior), for $R_q/Z > 1$ the charging energy is negligible (so-called superconducting behavior). We emphasize that our theoretical analysis predicts a singular point for $Z = R_q$, independently of the bare ratio $E_J/E_C$, and that the compact or extended nature of the junction phase does not play a role here.

## Exact diagonalization

To complete our investigation and fully characterize the emergence of the two phases we have applied exact diagonalization techniques[44,45] to our Hamiltonian (1) for finite-size systems. To extend the reachable sizes, we have taken full advantage of Josephson potential periodicity and of Bloch theorem. Indeed, we can introduce the eigenstates of the junction Hamiltonian $|\nu, s\rangle = e^{i\nu\varphi} u_\nu^s(\varphi)$, where $\nu$ is the quasi-charge and $s$ is the band index. In our charge gauge representation, the full

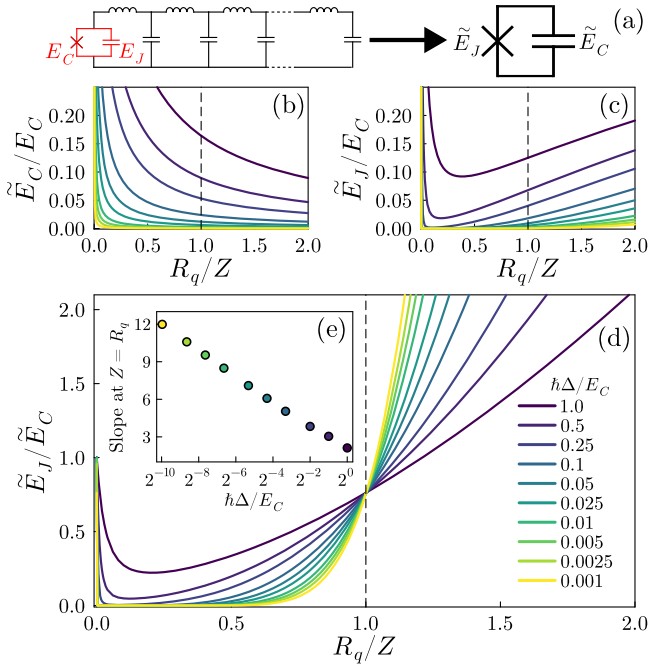

**Fig. 2 | Renormalization of the junction parameters. a** In the thermodynamic limit the full system can be understood as a renormalized junction.
**b, c** Renormalized capacitive and Josephson energies computed from (4) and (3) for different values of $\Delta$. **d** Ratio of the renormalized Josephson and charging energies. **e** Slope of the ratio at $Z = R_q$ showing logarithmic growth in the thermodynamic limit. For these plots $E_J = E_C$ and $\hbar\omega_p = 2E_C$.

Hamiltonian is block diagonal and for each $\nu$ we need to diagonalize the following matrix:

$$
\hat{H}_\nu = \sum_s \varepsilon_\nu^s |\nu, s\rangle\langle\nu, s| + \sum_{k=1}^{N_m} \hbar\omega_k \hat{a}_k^\dagger \hat{a}_k
$$
$$
+ i\sum_{k=1}^{N_m} g_k(\hat{a}_k^\dagger - \hat{a}_k) \sum_{s,r} \langle\nu, r|\hat{N}|\nu, s\rangle |\nu, r\rangle\langle\nu, s|,
$$

(5)

where $\varepsilon_\nu^s$ are the eigenenergies of the bare junction with $\nu \in [-0.5, 0.5]$ (Brillouin zone). Note that the spectrum is even with respect to $\nu$, so we can restrict to positive values only.

Illustrative energy bands for the bare junction are reported in Fig. 1b. An example of the obtained energy bands for the interacting system is shown in Fig. 3 versus $Z$. Different colors correspond to different values of the free spectral range for the same $\omega_p$ (i.e., to different numbers of modes), and the energies are rescaled by $\hbar\Delta$. For $R_q/Z \to 0$, the junction and the modes are decoupled: the spectrum is given by the bare junction bands with replicas due to a finite number of bosons in the modes. For finite $R_q/Z$ instead, the interaction manifests in energy anticrossings. To investigate a quantum phase transition, the behavior of the ground state and of the low-lying excited states is crucial. While increasing $R_q/Z$, the first energy band becomes narrower, corresponding to a reduced effective charging energy (see Fig. 2b). However, the curvature of the first band cannot change, since the sum rule (2) implies that $\widetilde{E}_C \geq 0$. This means that the ground state is always at $\nu = 0$. The first excited band instead changes its shape while varying $Z$. When the system is not coupled ($R_q/Z = 0$), it is simply a one-photon replica of the first band. Instead, when $R_q/Z$ is increased, the slope changes (see Fig. 3f, g). The low-energy spectrum of the full system at low impedances is reminiscent of the bare junction spectrum, although at much smaller energies. This qualitative behavior of the first few bands occurs for different sizes, although at smaller energies for smaller $\Delta$ (different colors in Fig. 3). Interestingly, the rescaled bands overlap in the best way for $Z = R_q$, confirming the universality observed in Fig. 2d. Away from this point, for both larger and smaller values of $Z$, the rescaled spectra do not overlap any longer. This is reminiscent of the finite-size scaling of continuous phase transitions.

## Behavior of ground and excited states

In the exact diagonalization studies, we considered an extended phase and applied Bloch's theorem. Now, since the full Hamiltonian for our finite-size system does not couple different quasi-charges $\nu$, the results for a compact phase are independent of the phase being fundamentally extended or compact. Indeed, we found that the ground state always occurs for $\nu = 0$, which corresponds to a periodic wavefunction. This independence of the phase transition on the extended/compact nature of the junction phase was also discussed

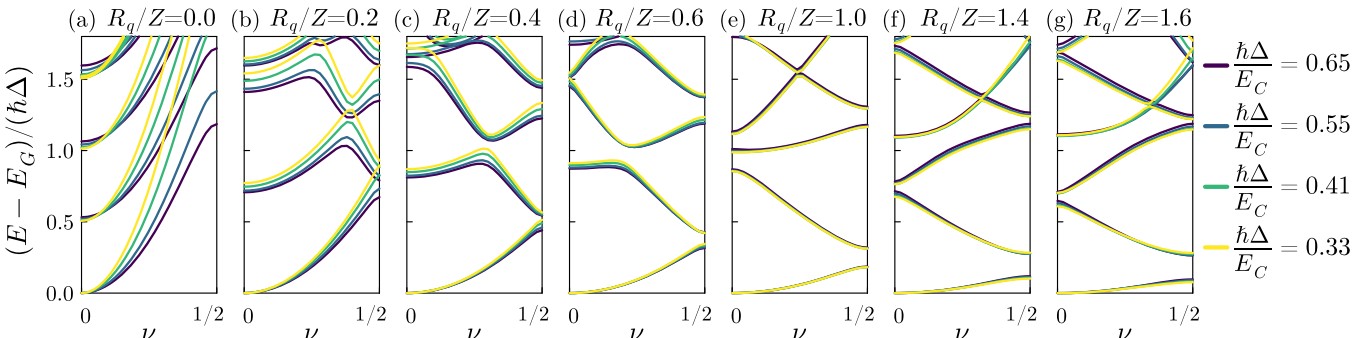

**Fig. 3 | Bands of the full system obtained by exact diagonalizaton. a–g** Bands for different impedances $Z$, as indicated in the subplot titles. The different line colors correspond to different free spectral ranges (see legend). The differences of the energies and the ground state energy $E_G$ are plotted and the vertical axis is rescaled by $\hbar\Delta$. In these plots $E_J = 0.5E_C$ and $\hbar\omega_p = 2E_C$.

in[22]. For more general shunts (e.g., inductive) an extended phase is relevant[27] and this may be needed to describe transport through the junction.

In Fig. 4a we plot the ground state energy versus $R_q/Z$: surprisingly, it has a weak dependence on the system size, and no precursor of a critical behavior is apparent. Figure 4b instead reports the energies of the first few excited states, separated from the ground state by an energy of the order of $\hbar\Delta$ for all the values of $Z$. In the thermodynamic limit, the system is expected to be gapless for all values of $Z$: hence, the phase transition needs to emerge with a different mechanism. Indeed, around $Z = R_q$ the first three excited levels exhibit an anticrossing. This corresponds to the observation we already made for Fig. 3 that the first excited band passes from being a one-photon replica to a dressed Josephson band. For a compact phase, this occurs around $Z = R_q$, with the splitting of the resulting anticrossing (but not the position) depending on $E_J$, as displayed in Fig. 5a. As in Fig. 3, while decreasing $\Delta$ all the rescaled spectra exhibit a universal behavior at $Z = R_q$. Hence, when approaching the thermodynamic limit, for a fixed $E_J$ the anticrossing becomes narrower and narrower, meaning that the signature of the phase transition first emerges in the first excited states. At the same time, the spectrum is gapless in the $\Delta \to 0$ limit so that a singular behavior in the first excited state necessarily reflects on the ground state.

While a compact phase is the only relevant quantity for the ground state of the system, the response of the system to external perturbations can depend on the full bands. To better highlight the behavior of the low-lying bands, in Fig. 4d we show the behavior of the gap between the first and second bands at the edge of the Brillouin zone. The ratio of the gap and the free spectral range displays a behavior analogous to the renormalized parameters of Fig. 2, with the gap closing for larger systems for $R_q/Z < 1$. This means that the low energy spectrum is approaching a capacitive "free particle" behavior. The opposite is true for $R_q/Z > 1$. As shown in Fig. 5b, this behavior extends to larger values of $E_J$.

Importantly, however, the ground state does not behave in the same way, as can be seen for example by computing observables for the junction degrees of freedom. As an example, we show in Fig. 4c the charge fluctuations $\sigma_N^2 = \mathrm{tr}(\hat{\rho}_J \hat{N}^2) - \mathrm{tr}(\hat{\rho}_J \hat{N})^2$, with $\hat{\rho}_J$ the reduced density matrix for the junction (analogous results are obtained, for example, for $\mathrm{tr}(\hat{\rho}_J \cos\hat{\varphi})$). For $R_q/Z > 1$ the charge fluctuations have a strong dependence on the system size. For $R_q/Z < 1$ instead, the size dependence is much smaller, and the fluctuations never fall below the Cooper pair box value that is obtained for $R_q/Z \to 0$ (horizontal dashed line). This, however, does not mean that the transition does not affect the ground state, as signaled by the different size dependence of the charge fluctuations on the two sides of $R_q/Z = 1$. The emergence of the singularity in the ground state in the thermodynamic limit is, however, slow, and we can only see a precursor.

We believe that this peculiar difference of behavior between the ground state and the excited states is at the origin of much of the recent controversy over this phase transition, specifically over the characterization of the so-called insulating state. In particular, in the thermodynamic limit, the ground state is not approaching a purely capacitive behavior. However, the response of the system to a gate charge (i.e., to a change in the quasi-charge) changes sharply at the transition point.

## Effect on higher frequency modes and spectroscopy

To understand the fate of the higher frequency modes at the transition point, we focus on the compact-phase Hamiltonian, given by equation (5) for $\nu = 0$. For $E_J = 0$, the dressed excited bands (charge states) energies are $\varepsilon_0^n(E_J = 0) = 4\tilde{E}_C n^2$, with $n$ an integer. In this limit, the effect of the transition on the photons can be understood by simply plotting the $Z$-dependent mode energies and the same energies shifted by the first dressed charge state $\varepsilon_0^1(E_J = 0)$. We show these two sets of energy branches in Fig. 6. The two lowest solid and dashed curves are

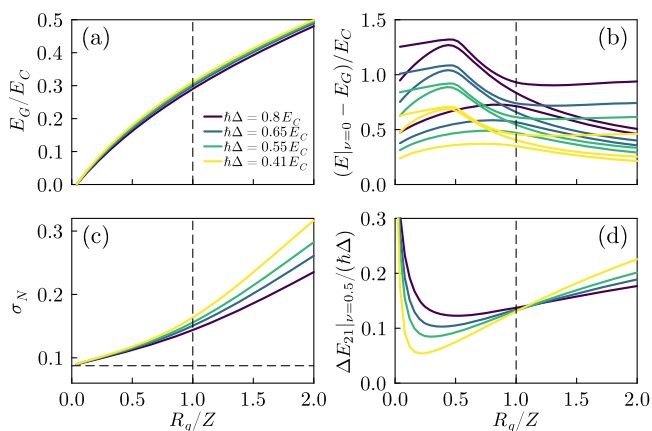

**Fig. 4 | Exact diagonalization results for different quantities. a** Ground state energy for different free spectral ranges. **b** Corresponding dependence of the first few excited state energies for a compact phase ($\nu = 0$). **c** Junction charge fluctuations in the ground state for different free spectral ranges. **d** Energy difference between the first two bands at the edge of the Brillouin zone $\nu = 0.5$. In all the panels $E_J = 0.5E_C$ and $\hbar\omega_P = 2E_C$.

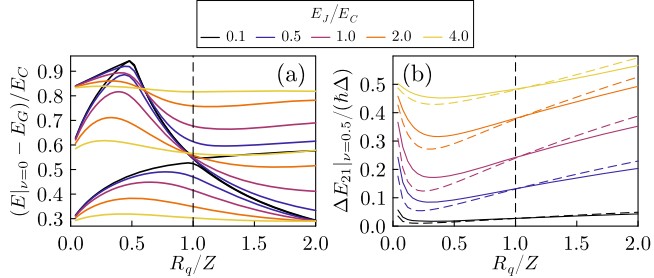

**Fig. 5 | Dependence of the spectra on the Josephson energy. a** Dependence on $E_J$ of the anticrossing involving the first three excited states for a compact phase ($\nu = 0$). Here $\hbar\Delta = 0.55E_C$ and $\hbar\omega_P = 2E_C$. **b** Energy difference between the first two bands at the edge of the Brillouin zone ($\nu = 0.5$) showing the universality at $R_q/Z = 1$ for a wide range of $E_J$. Solid lines $\hbar\Delta = 0.55E_C$, dashed lines $\hbar\Delta = 0.41$.

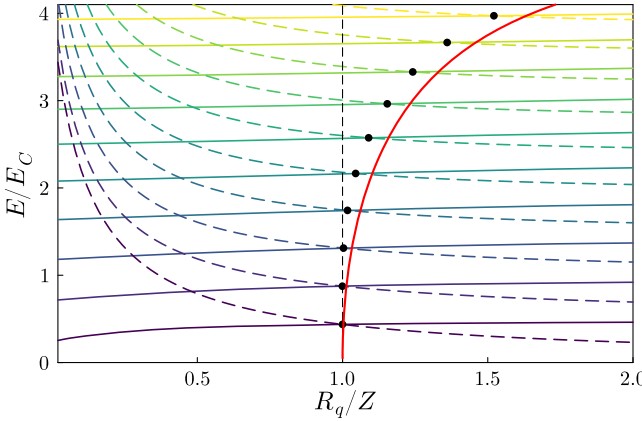

**Fig. 6 | Levels crossings involving the single-photon energies.** Single photon energies (solid lines) and the same energies shifted by the energy of the first excited renormalized junction band for $\nu = 0$ (dashed lines) with $E_J \to 0$, $\hbar\Delta = 0.5E_C$ and $\hbar\omega_P = 5E_C$. The $n$th mode energy and the $n$th level of the shifted branch have the same color. Their intersection point is highlighted by a black dot. The solid red line shows the intersections for a much smaller free spectral range $\hbar\Delta = 0.02E_C$.

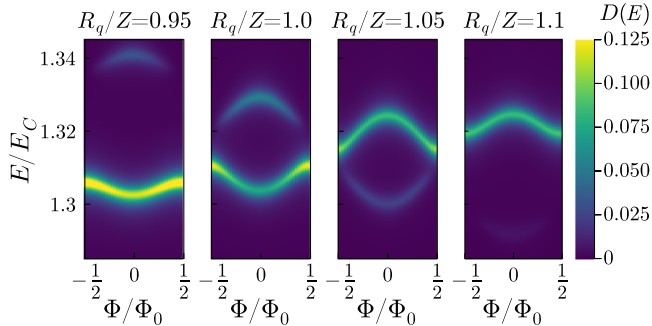

**Fig. 7 | Effect of the transition on the photonic spectral function (6).** Color plot of $D(E)$ around the bare energy of a photonic mode versus normalized flux bias ($\Phi_0 = h/2e$). The panels correspond to different values of $R_q/Z$ with $E_J = 0.1E_C$, $\hbar\Delta = 0.5E_C$ and $\hbar\omega_p = 5E_C$.

respectively the first photon energy $\hbar\omega_1$ and $\varepsilon_0^1(E_J = 0)$. Note that they cross at $Z = R_q$. With a finite $E_J$ the crossing is replaced by the anticrossing highlighted in Fig. 5a. Above these levels, for energies small enough with respect to $\hbar\omega_p$, also the $k$th photonic mode energy $\hbar\omega_k$ crosses the energy $\varepsilon_0^1(E_J = 0) + \hbar\omega_{k-1}$ at $Z = R_q$. This degeneracy is again lifted for $E_J \neq 0$. Hence, around the transition point, all the single photon states will hybridize. Note that the crossing of the first two curves, which is essentially determined by the universality highlighted in Fig. 2, is robust with respect to the intrinsic ultraviolet cutoff given by the Josephson plasma frequency $\omega_p$. For the excited states, there is eventually an important shift of the crossing point at high energies, that would be absent in the limit $\omega_p \to \infty$ (a non-dispersive line). The same shift is also present for much smaller free spectral ranges, of the order of the ones in the experiment[39], as shown by the red line in Fig. 6. For modes above the third, also states involving higher dressed bands play a significant role. Moreover, states with more than one photon are also present. For energies small enough with respect to $\hbar\omega_p$ (equidistant modes) these are also degenerate with the single-photon energies at the transition point, increasing the number of levels that participate the hybridization.

Recent experiments have observed a striking signature of the transition from the dispersion of high-frequency modes[39]. With the results of our exact diagonalization, we can calculate the linear-response photonic spectral function

$$D(E) = \sum_n \frac{\gamma^2}{\gamma^2 + (E - E_n + E_G)^2} |\langle G| \sum_k (a_k + a_k^\dagger)|E_n\rangle|^2, \quad (6)$$

where $E_n$ ($|E_n\rangle$) is the $n$th excited eigenenergy (eigenstate) of the full system, $E_G$ ($|G\rangle$) the ground energy (state) and $\gamma$ is a phenomenological broadening. Here, as in the experiment, we have considered a SQUID that is equivalent to a junction with $E_J$ that can be tuned by a magnetic field. Illustrative spectra versus the external magnetic flux $\Phi$ are shown in Fig. 7. We observe a clear change of sign of the photon frequency dispersion across the transition, as observed in[39]. This effect was linked to a capacitive/inductive behavior of the junction and here we see that it microscopically originates from the crossings observed in Fig. 6. The broadening observed in[39] for high energy modes is instead not present because of the moderate size of the system simulated here with exact diagonalization techniques. In fact, it relies on multiple resonances between single and multi-photon states[37]. In Fig. 7, one can see the state belonging to the shifted set of levels acquiring a finite single-photon component near the crossing point for $\Phi/\Phi_0 \neq \pm 0.5$, that is for $E_J \neq 0$. For larger and smaller impedances, this state is dark, i.e., it does not contribute to this spectral function.

## Discussion

In conclusion, we have revealed through an original analytical approach and exact diagonalization the emergence of the phase transition for a Josephson junction induced by the coupling to a high-impedance environment represented by a multimode transmission line. Our exact approach confirms hence the existence of the debated Schmid-Bulgadaev phase transition and shows that many distinctive features are already present with a moderate number of modes in a regime where the standard continuous models cannot be applied. Our study has evidenced unconventional properties of such phase transition and distinctive signatures on high-frequency spectroscopy. In particular we showed that for $Z < R_q$ the photonic frequencies experience a positive shift (inductive behavior) and for $Z > R_q$ a negative one (capacitive behavior), as evidenced in[39]. In particular, we have shown how the measured phase shift of the microwave reflection coefficient emerges from the microscopic energy level spectrum: this change of behavior occurs for all excited states at a small enough frequency with respect to the dispersive scale of the resonator.

Notice that the subtle connection with the superconducting/insulating transport properties of the present phases is not so obvious since this requires opening the system by coupling it to another circuit, a problem that needs to be addressed in the future. The universality we observe at the critical point is consistent with the standard theory of this phase transition, predicting from linear perturbation theory that for $Z > R_q$ the system will develop a potential drop when biased with a zero-frequency current source. Still, when biasing the system, the different quasi charges will not be independent anymore (the phase decompactifies). This implies that the excited states will play a role. The current disagreement over the so-called insulating state may be explained by the different behavior of the ground state and the excited states we evidenced, i.e., the fact that in the ground state (compact phase), the junction reduced density matrix does not approach a purely capacitive behavior.

The present work also paves the way to investigate emergent quantum phase transitions in complex environments with arbitrary transmission lines that can be engineered in a controlled way.

## Methods
### Derivation of the circuit QED Hamiltonian
We detail here the intermediate steps in the derivation of the Hamiltonian corresponding to the circuit of Fig. 1a, which can be described in terms of the independent flux node variables[41] included in the vector $\boldsymbol{\varphi}$. The procedure is analogous to the one used in[42]. The first step is the circuit Lagrangian

$$\mathcal{L} = \frac{1}{2}\frac{\hbar^2}{8E_C}\dot{\boldsymbol{\varphi}}^T \mathbf{C}\dot{\boldsymbol{\varphi}} - \frac{1}{2}\frac{\hbar^2}{8E_C}\boldsymbol{\varphi}\boldsymbol{\Gamma}\boldsymbol{\varphi} + E_J \cos\varphi_0, \quad (7)$$

where $\mathbf{C}$ and $\boldsymbol{\Gamma}$ are the capacitive and inductive matrices. Note that the pre-factors have been introduced to make $\mathbf{C}$ and $\boldsymbol{\varphi}$ dimensionless. Their expressions read:

$$\mathbf{C} = \frac{2E_C}{e^2}\begin{pmatrix} C_J & & & \\ & C & & \\ & & \ddots & \\ & & & C \end{pmatrix}, \boldsymbol{\Gamma} = \frac{2E_C}{e^2}\frac{1}{L}\begin{pmatrix} 1 & -1 & & & \\ -1 & 2 & -1 & & \\ & -1 & \ddots & \ddots & \\ & & \ddots & \ddots & -1 \\ & & & -1 & 1 \end{pmatrix}. \quad (8)$$

The Josephson junction charging energy is given in terms of its capacitance $C_J$ by $E_C = e^2/2C_J$.

The second step is to re-express such Lagrangian in terms of the normal modes. To get this, we perform a simultaneous diagonalization of the **C** and **Γ** matrices, given by a change of basis $\boldsymbol{\varphi} = \boldsymbol{P}\boldsymbol{\phi}$ that satisfies $\boldsymbol{\Gamma P} = \boldsymbol{CP}\boldsymbol{\omega}^2$, with $\boldsymbol{\omega}$ a diagonal matrix. The Lagrangian can therefore be written as

$$\mathcal{L} = \frac{1}{2}\frac{\hbar^2}{8E_C}\sum_{k=0}^{N_m}\left(\dot{\phi}_k^2 - \omega_k^2\phi_k^2\right) + E_J\cos\left(\sum_{k=1}^{N_m}P_{0k}\phi_k\right). \quad (9)$$

Introducing conjugate momenta (charges) $N_k = \partial\mathcal{L}/\partial\dot{\phi}_k$, the third step is to perform a Legendre transformation to get to the Hamiltonian form. It is convenient to change variables so as to identify the argument of the cosine as the junction phase $\varphi = \sum_k P_{0k}\phi_k$. This requires a redefinition of the junction charge $N = N_0/P_{00}$ and of the mode charges $n_k = N_k - P_{0k}N$. After this fourth step, we get the Hamiltonian

$$\mathcal{H} = 4E_C\left(P_{00}^2 + \sum_{k=1}^{N_m}P_{0k}^2\right)N^2 - E_J\cos\varphi + 4E_C\sum_{k=1}^{N_m}\left(n_k^2 + \left(\frac{\hbar\omega_k}{8E_C}\right)^2\phi_k^2\right)$$
$$+ 8E_C N\sum_{k=1}^{N_m}P_{0k}n_k. \quad (10)$$

By performing a canonical quantization of the Hamiltonian variables, by introducing bosonic creation and annihilation operators for the modes, and using the sum rule (2), one finally obtains the quantum Hamiltonian (1).

## Gauges and domain of the junction phase

We provide here additional details and remarks about the choice of the circuit gauge and the domain (extended versus compact) of the Josephson junction phase. With the diagonalization procedure detailed in the previous section, the Hamiltonian is in the so-called charge gauge because the transmission line operators are coupled to the junction via its charge operator. This gauge is analogous to the Coulomb gauge in cavity QED. Notice, however, the absence of a diamagnetic term that has been already implicitly eliminated in the diagonalization procedure. This is equivalent to starting with the diamagnetic term and diagonalizing the modes with a multi-mode Bogoliubov transformation. One could also consider a flux (dipole) gauge Hamiltonian, that can be obtained with a partial diagonalization of the Lagrangian or with a unitary transformation from the charge gauge one. In the flux gauge, the coupling between the junction and the modes is through the junction phase $\hat{\varphi}$. In this work, we have chosen to work with the charge gauge Hamiltonian because it is block diagonal once we apply the Bloch theorem.

Notice that a pure charge gauge Hamiltonian is possible if the first eigenvalue coming from the Lagrangian diagonalization is zero. This occurs only if the boundary condition at the end of the array is chosen to be an open one. This is due to the fact that the circuit we consider has de facto a superconducting island in the upper part of the array. If one also had, for example, an extra inductance in parallel to the whole array, the charge in the island would no longer be conserved and additional terms coupling to the flux of the junction would appear in equation (10).

This subtle point is related to the discussion on the domain over which the junction phase should take values[22,28]. In principle, values of the phase differing by $2\pi$ should correspond to the same physical state, and hence $\varphi$ should be a compact variable over one period of the cosine: this corresponds to a quantized conserved charge across the junction. However, once the junction is coupled to another circuit, this compactness is no longer obvious (notice also that we needed to take a linear combination of variables to identify the phase in the Hamiltonian (10)). Many treatments, including the ones predicting the Schmid-Bulgadaev transition, consider it to be extended over the whole real axis. In Ref. [19] it was argued that the transition is an incorrect prediction

deriving from not considering a compact phase. In our work, we have demonstrated how the phase transition emerges both for a compact and extended phase.

## Renormalization of the junction

We detail here some intermediate technical steps about the derivation of the renormalized Josephson energy and capacitive energy due to the coupling of the junction to the multi-mode transmission line. Let us consider the matrix elements of the Hamiltonian (1) expressed in the basis diagonalizing the sum of the last two terms of the Hamiltonian. For each mode, one can introduce a shifted bosonic operator $\hat{b}_k = \hat{a}_k + i\hat{N}\frac{g_k}{\hbar\omega_k}$ such that

$$\hbar\omega_k\hat{a}_k^\dagger\hat{a}_k + i\hat{N}g_k(\hat{a}_k^\dagger - \hat{a}_k) = \hbar\omega_k\hat{b}_k^\dagger\hat{b}_k - \hat{N}^2\frac{g_k^2}{\hbar\omega_k}. \quad (11)$$

The eigenstates of this part of the Hamiltonian are the tensor product of a charge eigenstate for the junction and of number states for the displaced operators $\hat{b}_k$. Note that these are displaced number states for the original operators $\hat{a}_k$, namely the states $|N,\mathbf{n}\rangle = |N\rangle\bigotimes_{k=1}^{N_m}|\alpha_{N,k}, n_k\rangle$ already introduced in the main text.

The Hamiltonian matrix elements between these states are hence

$$\langle M,\mathbf{m}|\hat{\mathcal{H}}|N,\mathbf{n}\rangle = \delta(M-N)\delta_{\mathbf{m,n}}\sum_k\hbar\omega_k n_k$$
$$+ \langle M,\mathbf{m}|\hat{\mathcal{H}}_J|N,\mathbf{n}\rangle - \delta(M-N)\delta_{\mathbf{m,n}}\hat{N}^2\sum_k\frac{g_k^2}{\hbar\omega_k}, \quad (12)$$

with the elements of the junction Hamiltonian being

$$\langle M,\mathbf{m}|\hat{\mathcal{H}}_J|N,\mathbf{n}\rangle = \langle M|\hat{\mathcal{H}}_J|N\rangle\prod_{k=1}^{N_m}\langle\alpha_{M,k}, m_k|\alpha_{N,k}, n_k\rangle. \quad (13)$$

The overlap of two displaced number states has the following expression, that can be obtained from the matrix elements of the displacement operator between number states[43],

$$\langle\beta, m|\alpha, n\rangle = e^{\frac{1}{2}(\alpha\beta^* - \alpha^*\beta)}e^{-\frac{1}{2}|\alpha-\beta|^2} \times \sqrt{\frac{n!}{m!}}(\alpha-\beta)^{(m-n)}L_n^{(m-n)}(|\alpha-\beta|^2), \quad (14)$$

for $m > n$, where $L_n^{(m)}$ are the associated Laguerre polynomials. The matrix elements (13) can hence be expressed as

$$\langle M,\mathbf{m}|\hat{\mathcal{H}}_J|N,\mathbf{n}\rangle$$
$$= \langle M|\hat{\mathcal{H}}_J|N\rangle e^{-\frac{|N-M|^2}{2}\sum_{k=1}^{N_m}\frac{g_k^2}{\hbar^2\omega_k^2}}\prod_{k=1}^{N_m}\sqrt{\frac{n_k!}{m_k!}}\left(\frac{g_k}{\hbar\omega_k}(N-M)\right)^{m_k-n_k}L_n^{(m_k-n_k)}\left(\frac{g_k^2}{\hbar^2\omega_k^2}|N-M|^2\right). \quad (15)$$

Notice that for $N = M$ the bare junction Hamiltonian matrix element is not modified. Hence the dressing of the junction by the transmission line only takes place for off-diagonal matrix elements in charge space, that is for the Josephson potential term.

Even if this expression is apparently complicated, there is an easily readable important effect on all the matrix elements. Since the Josephson potential can be also written as

$$E_J\cos\hat{\varphi} = \frac{E_J}{2}\int dN(|N\rangle\langle N+1| + |N+1\rangle\langle N|), \quad (16)$$

only matrix elements with $|N-M| = 1$ are nonzero. Hence, the exponential Franck-Condon-like factor before the product in equation (15) can be interpreted as giving the renormalized Josephson energy (3), independently from the number of photons in the different modes.

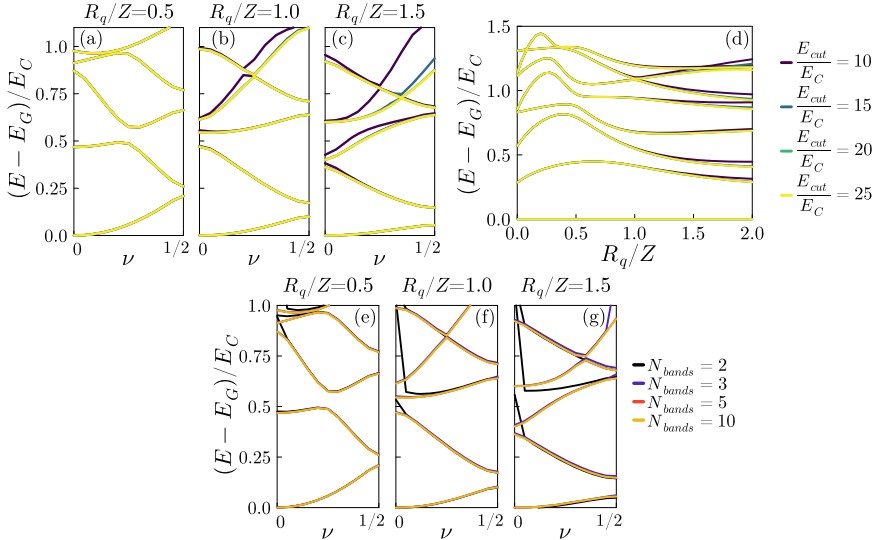

**Fig. 8 | Convergence of the exact diagonalization. a–d** Convergence of the spectra when increasing $E_{cut}$: **a** convergence for the full bands, **d** convergence for a compact phase ($\nu = 0$). Here $E_J = 0.5E_C$, $\hbar\omega_p = 2E_C$, $\hbar\Delta = 0.55E_C$, $N_{bands} = 5$. **e–g** Convergence of the bands when increasing the number of bare junction bands $N_{bands}$. Here $E_J = 0.5E_C$, $\hbar\omega_p = 2E_C$, $\hbar\Delta = 0.55E_C$, $E_{cut} = 15E_C$.

While the charging kinetic term of the junction is not directly affected by the dressing in (15), the last term of (12) proportional to $\hat{N}^2$ modifies it. This effect is analogous to the polaron dressing increasing the effective mass of a particle coupled to bosonic fields. In the present case, this corresponds to a decrease in the effective charging energy. Given the sum rule (2), the residual charging energy is given by equation (4).

Notice that while the sum appearing in the shift of the kinetic term is bounded because of the sum rule (2), the sum appearing in the exponential renormalizing of the Josephson energy (3) is not, and will diverge in the thermodynamic limit $\Delta \to 0$. This determines an exponential suppression of all the matrix elements (15), irrespective of the occupation numbers of the different modes. This is, therefore, the dominant effect in the thermodynamic limit.

### Details about the exact diagonalization

We report here some technical details about the exact diagonalization. All numerical computations were performed with the Julia programming language[45]. We have diagonalized the Hamiltonian (5) at fixed quasi-charge $\nu$, taking advantage of the fact that the full Hamiltonian is block diagonal in the quasi-charge. We represent (5) on the basis of eigenstates of the uncoupled system $\{|\nu, s\rangle \bigotimes_{k=1}^{N_m} |n_k\rangle\}$, where $s$ labels the bands and $|n_k\rangle$ is a Fock state of the $k$th mode. A finite basis is obtained by taking a fixed number of bands $s \in [1, N_{bands}]$ and by introducing an energy cutoff limiting the photonic states to the ones satisfying $\sum_j \hbar\omega_j n_j < E_{cut}$. Both $E_{cut}$ and $N_{bands}$ are numerical parameters that we need to take large enough to ensure convergence of the results. The convergence with respect to both parameters is shown in Fig. 8.

To determine the Hamiltonian matrix to diagonalize, we have first solved the Schrödinger equation for the bare junction using Bloch's theorem in order to obtain the bare energy spectrum and the matrix elements of the junction charge operator. Then, we numerically diagonalized the capacitive and inductive matrices to obtain the impedance-dependent frequencies of the transmission line modes and their coupling to the junction.

Since the size of the basis grows exponentially with the system size, in order to push to the limit the exact diagonalizations, it is important to have an efficient way of generating it given the energy cutoff constraint and to fill the (sparse) Hamiltonian. We have used numerical alogorithms similar in spirit to those described for example in[44]: in particular, we have generated the basis incrementally by

introducing a lexicographic order and filled the Hamiltonian by searching through ordered tags that uniquely identify each state. We have finally diagonalized the resulting sparse matrix by employing libraries that perform iterative solutions for computing the lowest eigenvalues.

Notice that the size of the basis increases quickly when increasing the number of modes and the numerical cutoff. The number of modes for which we performed exact diagonalization is, hence, moderate with respect to the ones considered in the computation of the renormalized Josephson and charging energies. It is interesting that the universality of the spectrum, which is expected to hold for large enough systems, is already well visible at these moderate sizes.

### Acknowledgements

We acknowledge support from the French project TRIANGLE (ANR-20-CE47-0011).

### Author contributions

L.G. and C.C. conceived the project, performed the theoretical analysis, interpreted the results, and wrote the manuscript. L.G. implemented and ran the codes.

### Competing interests

The authors declare no competing interests.

### Data availability

All data are displayed in the main text and Methods. Additional data are available from the corresponding author upon request.

### Code availability

The code that supports the plots in the main text and Methods is available from the corresponding author upon request.

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
