## [Peer Review File · Nature Communications]

Reviewers' comments:

Reviewer #1 (Remarks to the Author):

In this paper, the authors study a model of Josephson junction coupled to a transmission line by analytical and numerical methods. In particular, the authors focus on the finite-size scaling of the phase transition.

As far as I can see, the methodology and considerations look scientifically sound, and the results seem interesting for scientists working on related problems. On the other hand, I do not find the results significant for wider communities.

As one of the main findings, the authors emphasize that the ground-state energy exhibits no apparent singularity at the transition, and the transition is suggested rather by the structure of the excited states. Although the observation is interesting, the phenomenon is not thoroughly discussed. I think the lack of the apparent singularity in the ground-state energy is perhaps not too surprising, considering:

1. The transition is related to boundary critical phenomena in 1+1 dimensional conformal field theory. Even if the singularity exists, it is likely to be small and might be difficult to observe.
2. The transition also has some similarity to the Berezinskii-Kosterlitz-Thouless transition, which is known to exhibit very weak singularity in the free energy (or the ground-state energy of a 1D quantum system).

(I tried to find papers discussing the ground-state energy for the problem; I couldn't find but it might be still possible that some prior works exist.) I feel that this work could become more valuable by discussing the ground-state energy in more detail in Renormalization Group (RG) framework. Similarly, the authors' approach of the reducing the system to an effective circuit looks like a variation of RG transformation. It would be interesting to compare the authors' approach to the existing RG studies.

Even in the current status I think the paper is worth publication to stimulate further studies, but in a more specialized journal.

Reviewer #2 (Remarks to the Author):

The paper addresses the controversial issue of the dissipative quantum phase transition (DQPT) in a single Josephson junction. While originally and in many following papers they considered the junction in ohmic environment, the authors consider the junction in the multimode transmission line

environment. Recently the latter case attracted great attention and interesting experiments have been done [39].

The theoretical analysis is comprehensive and looks reliable. I support publication of the paper. At the same time, I have a couple of comments for consideration by the authors:

1. There is a delicate problem of independence of the DQPT on E_J/E_C . The formal theory correctly predicts this independence indeed. But in a real experiment at very large E_J/E_C it is impossible to observe the transition from superconducting to insulating state because of inevitably finite temperature, observation time, or voltage measurement error. This was discussed for ohmic environment [18,22]. Are there similar restrictions on observation of DQPT in transmission line environment? Could the authors address this issue?
2. Independence of the DQPT on the choice of extended or compact phase was discussed in details in Ref. [23].

Reviewer #3 (Remarks to the Author):

The authors study a small Josephson junction shunted by a transmission line with an impedance Z of the order R_Q , the superconducting conductance quantum. Their approach is to replace the transmission line with a finite system and to study how physical properties of the system change as the size of the finite system is increased. In this way they hope to shed new light on the quantum phase transition known as the Schmid-Bulgadaev transition that should occur around $Z=R_Q$.

The study is motivated by what several independent authors have recently characterised as unresolved questions regarding this transition. As Houzet et al (arXiv:2308.16072) puts it, "[s]o far, the phase diagram experimentally inferred from the dc response of shunted Josephson devices, is far from reproducing the predicted phase diagram. Modern attempts to observe the Schmid transition rely on finite-frequency measurements [...] Much less [than about thermal fluctuations] is known on the role of a finite frequency that was mostly studied in perturbative regimes." While it is clear that one major open question is how to study the transition using finite frequency probes, a recent work which failed to find experimental signatures (reference 20 in the submitted manuscript) suggested that the problem may be with our theoretical understanding of the problem. This triggered comments, replies and theoretical papers, putting forth arguments for or against the correctness and applicability of the canonical theory, which had previously been considered uncontroversial. Masuki

et al (reference 24 in the submitted manuscript) suggested that the conventional theory does not apply when the charging energy is smaller than both the Josephson energy and the plasma frequency. While I believe Masuki et al's is incorrect (see reference 25 in the submitted manuscript), I think it is fair to say that there is lack of consensus as to whether the transition occurs at $Z=R_Q$ when E_J is larger than E_C . A study that resolves this theoretical question unambiguously would in my opinion warrant publication in Nature Communications.

The authors present the following significant results:

- 1) They employ a "polaron"-type unitary transformation to cast all but the Josephson term of the Hamiltonian in diagonal form, and from this read off renormalized Josephson and charging energies for the small junction. Finite size scaling indicates that the renormalized Josephson energy dominates the renormalized charging energy when $Z < R_Q$, and the other way round for $Z > R_Q$.
- 2) They exactly diagonalise truncated systems and study the obtained spectra as the system size is increased. They show indications that excited state level crossings and a universal spectral dependence on quasi-charge at $Z=R_Q$ emerge in the thermodynamic limit.

My assessment of the significance of these results are as follows. I think that it provides valuable intuitive insights into the Schmid transition: Perhaps the polaron picture that is behind the results in Figure 2 will stimulate future work. The point that is made in figure 5, namely that at small E_J/E_C , there is a level inversion between the modes of the resonator and the dressed junction frequencies at exactly R_Q , illuminates the Schmid transition from a perspective I have not thought of before. For these reasons, I think the work deserves to be published. However, I do not think that the work presents advances that are of the significance required to justify publication in Nature Communications.

I base this judgement on the following:

- 1) The results presented here and throughout the paper are for E_J/E_C between 0.1 and 1, and given the logarithmic scaling, the method probably cannot convincingly address the regime $E_J \sim 10 E_C$. To answer open questions about the transition would require a convincing calculation of the phase boundary up to at least $E_J = 5 E_C$.
- 2) The exact diagonalization results are again done at small E_J/E_C and therefore at best confirm the uncontroversial fact that at these E_J/E_C values, the transition occurs at $Z=R_Q$. However, there are further questions to ask: Can excited state level crossings be used to infer a ground state quantum phase transition? Is the level crossing between the second and third state really exactly at $Z=R_Q$. It seems from figure 2 that it may shift to smaller Z at larger bare E_J ? There seems to be another level crossing between states 3 and 4 at $Z=2 R_Q$ in figure 4c. What is the significance of that, if the crossing at $Z=R_Q$ is supposed to indicate the Schmid transition.

Overall I would say that the paper does not contain strong evidence for or against the proposition that the Schmid transition occurs at large $Z=R_Q$ when bare E_J/E_C is large, and therefore does not address any major open problem, but rather provides useful new perspectives on known results.

Some further issues:

3) The results in Figure 2 are very suggestive of a quantum phase transition, but falls short of a complete proof. A complete proof requires one to show that the observed behavior of renormalized E_J/E_C leads to a non-analyticity in the ground state as function of Z in the thermodynamic limit. Together with the fact that the calculations are not in the $E_J \gg E_C$ limit, I would say these results are weaker than the ones originally obtained by Schmid and others four decades ago. (They do have the virtue though, of giving a new perspective.)

4) In discussing the results of Figure 5, it is said that the level crossing occurs at $Z=R_Q$ for modes with frequencies sufficiently below the plasma frequency. However, I would expect that it is rather for modes below E_C . This is because the junction capacitance provides the UV short circuit that cuts off Josephson tunnelling when $E_C < \omega_p$. Indeed, deviations are seen at lower frequencies in the right panel of Figure 5, when E_C is reduced.

5) I am not sure the results in Figure 6 have much significance. Firstly, a demonstration of inductive/capacitive behavior on either side of R_Q of the junction is only consistent with the Schmidt transition, but not a smoking gun. Secondly, all high energy modes should bend down with flux increased from 0 to $\pi/2$ for $Z < R_Q$ or up for $Z > R_Q$, but the figure shows two modes bending in opposite direction at each Z . In general, I am sceptical that exact diagonalization of a bosonic system with a non-linearity can reach large enough systems and numbers of bosonic quanta to obtain results in qualitative (let alone quantitative) agreement with the behaviour of a Josephson junction shunted by a large many-mode resonator. To convince the reader, the authors should present a convergence analysis in the supplementary material to show that their exact diagonalization results are sufficiently converged as function of number of resonator modes, and number of bosonic quanta.

For these reasons, I recommend that the paper be rejected by Nature Communications, but hope that a revised version will be accepted in a more appropriate journal (such as for instance PRB).

Izak Snyman

Reply to Referee 1

Referee: *In this paper, the authors study a model of Josephson junction coupled to a transmission line by analytical and numerical methods. In particular, the authors focus on the finite-size scaling of the phase transition. As far as I can see, the methodology and considerations look scientifically sound, and the results seem interesting for scientists working on related problems. On the other hand, I do not find the results significant for wider communities.*

We appreciate the positive feedback from the referee regarding our manuscript. While we acknowledge the referee's view on the perceived significance for wider communities, we note that specific reasons for this viewpoint were not provided.

We firmly believe in the importance of our work as it contributes to the clarification of various aspects within a controversial and timely topic. Notably, our investigation reveals surprising universality behaviors within the system, which we believe could capture the interest of a diverse audience.

Referee: *As one of the main findings, the authors emphasize that the ground-state energy exhibits no apparent singularity at the transition, and the transition is suggested rather by the structure of the excited states. Although the observation is interesting, the phenomenon is not thoroughly discussed. I think the lack of the apparent singularity in the ground-state energy is perhaps not too surprising, considering: 1. The transition is related to boundary critical phenomena in 1+1 dimensional conformal field theory. Even if the singularity exists, it is likely to be small and might be difficult to observe.*

In the revised version of the manuscript we added a discussion and some plots to address the ground state properties of the system. Even if the slow development of a singularity in the ground state properties can be expected, we feel that the most interesting part of our results is in the emergent features, that mainly involve the excited states and the extended Bloch bands. These in fact show clear signs of a transition already at moderate system sizes, a result that is peculiar and nontrivial.

Referee: *2. The transition also has some similarity to the Berezinskii-Kosterlitz-Thouless transition, which is known to exhibit very weak singularity in the free energy (or the ground-state energy of a 1D quantum system). (I tried to find papers discussing the*

ground-state energy for the problem; I couldn't find but it might be still possible that some prior works exist.)

The referee has not quoted any references, so it is impossible to comment further on the invoked similarity. We also previously searched deeply in the literature for works discussing this point, but with no success. Once again, to the best of our knowledge, there is no study in the literature about the emergent of the transition with finite-size exact diagonalization approaches as the one in our manuscript.

Referee: *I feel that this work could become more valuable by discussing the ground-state energy in more detail in Renormalization Group (RG) framework. Similarly, the authors' approach of the reducing the system to an effective circuit looks like a variation of RG transformation. It would be interesting to compare the authors' approach to the existing RG studies.*

We acknowledge the referee's recommendation and recognize the potential merit in exploring the ground-state energy in more detail within the framework of Renormalization Group (RG). However, our chosen methodology in this study involves the application of the exact diagonalization method. This decision is motivated by recent research, particularly [Masuki et al. PRL (2022)], which questions the validity of initial predictions related to a vertical transition line. Additionally, the ensuing discussions critiquing Masuki et al.'s work ([Sépulcre et al. arXiv:2210.00742 (2022)], [Daviet, Dupuis arXiv:2307.04835 (2023)]) underscore the inherent complexities associated with employing Renormalization Group techniques to address this specific problem.

Reply to Referee 2

Referee: *The paper addresses the controversial issue of the dissipative quantum phase transition (DQPT) in a single Josephson junction. While originally and in many following papers they considered the junction in ohmic environment, the authors consider the junction in the multimode transmission line environment. Recently the latter case attracted great attention and interesting experiments have been done [39].*

The theoretical analysis is comprehensive and looks reliable. I support publication of the paper.

We thank the referee for the positive comments on our work and for suggesting publication.

Referee: *At the same time, I have a couple of comments for consideration by the authors:*

- 1. There is a delicate problem of independence of the DQPT on E_J/E_C . The formal theory correctly predicts this independence indeed. But in a real experiment at very large E_J/E_C it is impossible to observe the transition from superconducting to insulating state because of inevitably finite temperature, observation time, or voltage measurement error. This was discussed for ohmic environment [18,22]. Are there similar restrictions on observation of DQPT in transmission line environment? Could the authors address this issue?*

This is indeed a subtle point. The considered Schmid phase diagram is an equilibrium one. While the phase boundary is independent on the ratio E_J/E_C , reaching thermal equilibrium is not trivial in the "transmon" regime, when the wells of the Josephson cosine potential are *de facto* separated. In other words, if the tunneling between wells requires the age of the universe, in practice it is not possible to observe equilibrium physics. We think that this point extends to transmission line environments. Recent experiments [Kuzmin et al., arXiv 2023] have focused on the regime where E_J/E_C is of order one.

Referee: *2. Independence of the DQPT on the choice of extended or compact phase was discussed in details in Ref. [23].*

We thank the Referee pointing this out, we have made this point clearer in the revised manuscript.

Reply to Referee 3

Referee: *The authors present the following significant results: 1) They employ a "polaron"-type unitary transformation to cast all but the Josephson term of the Hamiltonian in diagonal form, and from this read off renormalized Josephson and charging energies for the small junction. Finite size scaling indicates that the renormalized Josephson energy dominates the renormalized charging energy when $Z < R_Q$, and the other way round for $Z > R_Q$. 2) They exactly diagonalise truncated systems and study the obtained spectra as the system size is increased. They show indications that excited state level crossings and a universal spectral dependence on quasi-charge at $Z = R_Q$ emerge in the thermodynamic limit.*

My assessment of the significance of these results are as follows. I think that it provides valuable intuitive insights into the Schmidt transition: Perhaps the polaron picture that is behind the results in Figure 2 will stimulate future work. The point that is made in figure 5, namely that at small E_J/E_C , there is a level inversion between the modes of the resonator and the dressed junction frequencies at exactly R_Q , illuminates the Schmid transition from a perspective I have not thought of before. For these reasons, I think the work deserves to be published.

We thank the Referee for having carefully read our work and for his positive assessment of our results. We are especially glad that he found some of our results illuminating.

Referee: *However, I do not think that the work presents advances that are of the significance required to justify publication in Nature Communications. I base this judgement on the following: 1) The results presented here and throughout the paper are for E_J/E_C between 0.1 and 1, and given the logarithmic scaling, the method probably cannot convincingly address the regime $E_J > 10E_C$. To answer open questions about the transition would require a convincing calculation of the phase boundary up to at least $E_J = 5E_C$.*

The referee rightly points out the challenges associated with addressing the regime of $E_J \sim 10E_C$ and the slow convergence of exact diagonalization. In the revised version, we have included additional results pertaining to the E_J dependence of the spectra for higher E_J , demonstrating emergent universality and characteristic anticrossings up to $E_J = 4E_C$ (new Figures 5(a)-(b), reported here as Fig.A).

FIG. A: (a) Dependence on the Josephson energy E_J of the anticrossing between the first few excited states for a compact phase. Here $\hbar\Delta = 0.55E_C$ and $\hbar\omega_p = 2E_C$. (b) Band gap at the edge of the Brillouin zone showing the universality at $R_q/Z = 1$ for a wide range of E_J . Solid lines $\Delta = 0.55E_C$, dashed lines $\Delta = 0.41$.

It is crucial to emphasize, however, that the E_J dependence of the transition is not the sole unresolved issue regarding this transition, nor is it the primary focus of our study. The very existence of the transition has been a subject of dispute [Murani et al. PRX (2020); Hakonen and Sonin PRX (2021); Murani et al. PRX (2021)], and the nature of the insulating state remains under debate. Additionally, while we do not endorse the findings of [Masuki et al. PRL (2022)], they underscore a critical concern about the applicability of existing theories, challenging the unambiguous occurrence of the transition at $Z = R_Q$.

Given these considerations, demonstrating the existence of the transition itself is a significant outcome of our work. While not constituting a formal proof, our finite-size-scaling unequivocally reveals the emergence of a capacitive regime at high impedances and underscores the universality of the critical point. Furthermore, we directly address the excited states, a departure from prior works.

Our study brings clarity to several outstanding issues related to this transition, including the role of the phase domain. We have introduced two panels in Fig. 4 and incorporated a discussion on the ground state of the system, revealing distinct behavior compared to the excited states. This distinction has the potential to shed light on the controversy surrounding the nature of the so-called insulating phase.

Referee: 2) *The exact diagonalization results are again done at small E_J/E_C and therefore at best confirm the uncontroversial fact that at these E_J/E_C values, the transition occurs*

at $Z = R_Q$. However, there are further questions to ask: Can excited state level crossings be used to infer a ground state quantum phase transition? Is the level crossing between the second and third state really exactly at $Z = R_Q$. It seems from figure 2 that it may shift to smaller Z at larger bare E_J ? There seems to be another level crossing between states 3 and 4 at $Z = 2R_Q$ in figure 4c. What is the significance of that, if the crossing at $Z = R_Q$ is supposed to indicate the Schmid transition.

As discussed earlier, the assertion of the transition's occurrence at small E_J/E_C is far from universally accepted. Even in the case of [Masuki et al. PRL (2022)], who predict its existence, it deviates from $Z = R_Q$ even at very low E_J/E_C .

The hallmark indication of the phase transition in our findings lies in the universality of the spectra at $Z = R_Q$, substantiated by the new band gap plots in Figures 4 and 5. This implies that, given sufficient system size, the response diverges markedly on either side of $Z = R_Q$. The anticrossings, illustrated in these figures, manifest this universality within the compact phase subspace, elucidating the capacitive/inductive shifts in the environmental modes. We specifically focus on the anticrossing involving the first excited state since, in the thermodynamic limit, it converges to the ground state, rendering the spectrum gapless. Given that a singularity in ground state properties arises from level crossings, the first excited state undergoes this critical crossing.

The broader question of deducing a ground state phase transition from excited states is pivotal. Our additional results in Figure 4 underscore the emergence of features in the excited states (or extended bands) preceding those in the ground state. This observation aligns with recent experimental findings [Kuzmin et al. arXiv:2304.05806 (2023)], affirming the distinct response of the system on either side of the transition. While the manifestation of singularity in ground state properties is gradual in this system, discernible size-dependent trends in observables on opposing sides of the transition (Fig. 4(c)) foreshadow this impending singularity.

Referee: *Overall I would say that the paper does not contain strong evidence for or against the proposition that the Schmid transition occurs at large $Z = R_Q$ when bare E_J/E_C is large, and therefore does not address any major open problem, but rather provides useful new perspectives on known results.*

Again, we disagree on the fact that the behaviour of the transition at large E_J/E_C is the

only major open problem about the Schmid transition.

Referee: *Some further issues: 3) The results in Figure 2 are very suggestive of a quantum phase transition, but falls short of a complete proof. A complete proof requires one to show that the observed behavior of renormalized E_J/E_C leads to a non-analiticity in the ground state as function of Z in the thermodynamic limit. Together with the fact that the calculations are not in the $E_J \gg E_C$ limit, I would say these results are weaker than the ones originally obtained by Schmid and others four decades ago. (They do have the virtue though, of giving a new perspective.)*

It is true that our calculations do not explore the $E_J \gg E_C$ limit, a deliberate choice influenced by recent discussions challenging the validity of predictions in that limit [Masuki et al.]. Our focus on moderate E_J/E_C aims to contribute to ongoing debates and provide a comprehensive analysis in a broader parameter space.

While we may fall short of a complete proof, our results introduce a fresh perspective and valuable insights into the system's behavior, particularly in areas of recent contention. We hope our findings will stimulate further investigation and discussion, recognizing the ongoing debates surrounding this complex topic.

Referee: *4) In discussing the results of Figure 5, it is said that the level crossing occurs at $Z = R_Q$ for modes with frequencies sufficiently below the plasma frequency. However, I would expect that it is rather for modes below E_C . This is because the junction capacitance provides the UV short circuit that cuts off Josephson tunnelling when $E_C < w_p$. Indeed, deviations are seen at lower frequencies in the right panel of Figure 5, when E_C is reduced.*

This is an interesting point. However, it seems that it is solely the plasma frequency that controls this behaviour. In fact, by repeating the calculation with the same mode spacing but a larger plasma frequency we see a straightening of the intersection line, as shown in the attached Figure B. A peculiarity of this result is that it does not depend on E_J , that only controls the width of the resulting anticrossing.

Referee: *5) I am not sure the results in Figure 6 have much significance. Firstly, a demonstration of inductive/capacitive behavior on either side of R_Q of the junction is only consistent with the Schmidt transition, but not a smoking gun. Secondly, all high energy modes should bend down with flux increased from 0 to $\pi/2$ for $Z < R_Q$ or up for $Z > R_Q$, but the figure shows two modes bending in opposite direction at each Z . In general, I am*

FIG. B: Dependence on the plasma frequency of the intersection between photonic energies and dressed junction levels

sceptical that exact diagonalization of a bosonic system with a non-linearity can reach large enough systems and numbers of bosonic quanta to obtain results in qualitative (let alone quantitative) agreement with the behaviour of a Josephson junction shunted by a large many-mode resonator.

We agree that the inductive/capacitive behaviour at finite frequency is only consistent with a ground state phase transition. We have made sure to clarify this point.

We however feel that it is important to show this result (now in Fig.7) because it still is an important part of the physics of the system and because it connects with recent experiments [Kuzmin et al. arXiv:2304.05806 (2023)].

While we do not expect to recover the full physics of a junction coupled to a very large number of modes this result shows that the observed shift in the photon frequencies originates in the crossings between the excited levels we highlighted in Fig.6 (old Fig.5) and that it is present already for a small number of modes.

More in general, we are convinced that our results highlight that many interesting features of a junction coupled to a multi-mode resonator are already present with a small number of modes.

For what concerns the two modes visible in Fig.7 (old Fig.6), we understand the potential source of confusion. One can see that the brightest line continuously connects to the bare mode frequency when $E_J = 0$ ($\Phi/\Phi_0 = 0.5$). This is the line showing the change in the mode frequency dispersion on the two sides of the transition. The other line is the corresponding dressed junction level (dashed lines in Fig.6(a) (old Fig.5(a))), that is dark away from the transition point, but that necessarily acquires a photonic component in the anticrossing

FIG. C: Additional plots of the change of the mode dispersion. Parameters as in Fig.7.

region. We show in Fig.C an extra plot to highlight this behaviour. In a system with many modes also resonances with multiphoton states will become important for this observable, causing a broadening of the lines.

We updated our discussion to clarify this point, we thank the referee for their feedback on the clarity of our exposition.

Referee: *To convince the reader, the authors should present a convergence analysis in the supplementary material to show that their exact diagonalization results are sufficiently converged as function of number of resonator modes, and number of bosonic quanta.*

The referee is right, it is important to systematically assure convergence of the results. We added an analysis in the supplementary material showing the convergence of our numerics.

Brief summary of changes

- We added one sentence in the abstract about the new added results on the characterisation of the phases
- We have improved the introduction to make it sharper
- We updated the discussion on the renormalized junction parameters
- We updated Fig. 3 with new data
- We updated Fig. 4
 - The ground state energy plotted in panel (a) did not include the zero point energy of the environmental modes is changing with impedance. We corrected this.
 - Panels (c) and (d) were replaced with different plots. Panel (c) now shows the impedance and size dependence of the junction charge fluctuations. Panel (d) shows the impedance and size dependence of the gap between the first two bands at the edge of the Brillouin zone.
- The plot showing the E_J dependence of the first few excited levels was moved to the newly added Fig.5(a). Curves for higher values of E_J/E_C were added.
- The newly added plot in Fig.5(b) shows the behaviour of the band gap at the edge of the Brillouin zone. This displays the universality at $Z = R_q$ also at higher values of E_J .
- Correspondingly to these changes in the figures, we added a discussion about the behaviour of the ground state and the excited states.
- We added comments to the results reported in Fig.6.
- In the Methods we added a plot and a discussion showing explicitly the convergence of our numerical computations.

REVIEWER COMMENTS

Reviewer #3 (Remarks to the Author):

In my original report, I stated that the submitted manuscript “does not address any major open problem, but rather provides useful new perspectives on known results.” The revised manuscript does not contain any significant extensions of the theory or results presented in the original manuscript.

Rather, the authors deal with my assessment in their rebuttal, arguing as follows. (I quote from their rebuttal letter.)

- 1) “It is crucial to emphasize, however, that the EJ dependence of the transition is not the sole unresolved issue regarding this transition, nor is it the primary focus of our study. The very existence of the transition has been a subject of dispute [Murani et al. PRX (2020); Hakonen and Sonin PRX (2021); Murani et al. PRX (2021)], and the nature of the insulating state remains under debate.”
- 2) “Given these considerations, demonstrating the existence of the transition itself is a significant outcome of our work.”
- 3) “As discussed earlier, the assertion of the transition’s occurrence at small EJ/EC is far from universally accepted.”

This reveals the fundamental disagreement between me and the authors.

My position is that the existence of the transition is not under dispute, and that there is no controversy regarding the small EJ/EC regime. I am not aware that any errors have been discovered in the body of work on this topic over the past four decades that would require us to rewrite our textbooks. The work of Murani et al. that the authors base their rebuttal on, comprises a finite frequency measurement of a single sample. The comment of Hakonen and Sonin that the authors cite, clarifies why this does not contradict the established theory. My take on the questions raised by Murani et al.’s work is summarized in the statement by Houzet et al. that I quoted in my initial report, namely that there are experimental challenges that prevent full comparison of the theory for the DC response to experiment, and an incomplete understanding of how to study the effect using finite frequency probes.

My assessment of the work remains that it provides circumstantial evidence for the validity of the conventional picture of the Schmidt transition at small E_J/E_C , and does not address the more challenging large E_J/E_C regime or answer any major open questions. My recommendation remains not to accept the paper in Nature Communications.

I have one further specific comment regarding the data at $E_J/E_C=4.0$ in Figure 5 (a): If there is an anticrossing that leads to a phase transition, then $(E_4-E_3)/\Delta$ and $(E_3-E_2)/\Delta$ should go to zero at $Z=R_Q$ when $\Delta \rightarrow 0$, i.e. the first three levels should become degenerate on the scale of Δ as Δ decreases. There is no suggestion of such an anticrossing at $E_J/E_C=4.0$, and this suggests that the authors are incorrect to claim that the anticrossing is a general feature of the phase transition.

Reviewer #4 (Remarks to the Author):

The authors address the important problem of Schmid phase transition in a Josephson junction coupled to the environment. By obtaining the exact numerical solution for a system with the junction coupled to a transmission line, they confirm that the phase transition occurs at the critical value of the transmission line impedance, $Z=R_Q$. Namely, the authors demonstrate that at $Z=R_Q$ the avoided crossing between the second and the third excited states of the system is happening, and they relate this feature to the phase transition.

I am confident that the reported results are formally correct.

Indeed, the authors carefully analyze the Hamiltonian of the system without introducing artificial cutoffs etc., as it is often done.

Therefore, their results can be potentially used to analyze real microwave experiments, as they themselves demonstrate in Fig. 7, comparing their numerics to the experiment [40]. For example,

I believe that it is possible to verify the spectra plotted in Fig. 3 experimentally. Therefore, in my personal opinion Fig. 3 is the main result of the paper. I also find very useful the analysis of the renormalized Josephson and charging energies shown in Fig. 2 a,b. These plots clearly demonstrate that a very long transmission line is not fully equivalent to an Ohmic resistor, as it is sometimes believed, because it adds a big shunting capacitor to the junction. At the same time, in the standard formulation of the problem of a Josephson junction shunted by Ohmic resistor the counter term is introduced to compensate for the renormalization of the charging energy. For these reasons I recommend accepting the paper.

I believe, however, that the presentation of the results can be improved. It would be useful if the authors would extend the discussion of their results on a qualitative level because currently the paper is written in a too formal way. For example, it is not clear what are the physical consequences of the phase transition found by the authors. Namely, how does the avoided crossing between the third and the second excited states influence the microwave reflection coefficient, for example? From the discussion of Fig. 7 in the text I understood that the phase transition implies the switching from the capacitive to the inductive response at low frequencies. It is a very clear and intuitive picture of the phase transition. However, in the conclusions the authors state that this issue is very subtle and sensitive to the circuit design etc. Can one say something definite strictly for the circuit shown in Fig. 1a? Does the junction in this circuit act as a capacitor in the insulating phase and as an inductor in the superconducting phase if the transmission line is sufficiently long?

REPLY TO REFEREE 3

Referee: *My position is that the existence of the transition is not under dispute, and that there is no controversy regarding the small E_J/E_C regime. I am not aware that any errors have been discovered in the body of work on this topic over the past four decades that would require us to rewrite our textbooks. The work of Murani et al. that the authors base their rebuttal on, comprises a finite frequency measurement of a single sample.*

(...)

My assessment of the work remains that it provides circumstantial evidence for the validity of the conventional picture of the Schmidt transition at small E_J/E_C , and does not address the more challenging large E_J/E_C regime or answer any major open questions.

We thank Referee 3 for evaluating our revised manuscript and for providing his insight. Recent papers on the subject witness that there is a debate, also on the small E_J/E_C regime. In fact a very recent theory preprint even claims the non-existence of the transition (arxiv.org/abs/2312.14754).

The fact that there is a debate is not based only on the work by Murani et al. (PRX 2020), but also on the letter by Masuki et al (PRL2022). We are aware that the Referee 3 expressed doubts on the validity of their numerical renormalization group procedure, but to the best of our knowledge their arguments criticizing the validity of the models used to derive the transition has not been disputed. The main goal of our work is exactly to address the doubts on the effective models used in the past and to show the emergence of the transition with no approximations. To our knowledge we are the first to do that and to show where the transition originates.

Referee: *I have one further specific comment regarding the data at $E_J/E_C = 4.0$ in Figure 5 (a): If there is an anticrossing that leads to a phase transition, then $(E_4 - E_3)/\Delta$ and $(E_3 - E_2)/\Delta$ should go to zero at $Z = R_Q$ when $\Delta \rightarrow 0$, i.e. the first three levels should become degenerate on the scale of Delta as Delta decreases. There is no suggestion of such an anticrossing at $E_J/E_C = 4.0$, and this suggests that the authors are incorrect to claim that the anticrossing is a general feature of the phase transition.*

We thank Referee 3 for this remark that helped us to improve our discussion of Fig.5. The universality of the energy levels divided by Δ at $Z = R_Q$ holds irrespective of E_J/E_C , as shown in Fig. 5b. In Fig. 5a, we show transition energies normalized E_C for a fixed

Δ . It is therefore not surprising that for different values of E_J/E_C the finite-size effects are different, in particular more pronounced for the more challenging regime of high E_J/E_C values. To make this point clearer, we have introduced a footnote in the revised manuscript.

REPLY TO REFEREE 4

We thank Referee 4 for the positive evaluation of our work and for finding it interesting and of experimental relevance. We particularly appreciate that the difference of our exact approach with respect to previous studies has been explicitly acknowledged by Referee 4.

We are also grateful for the constructive remarks to improve the manuscript. In what follows we carefully address the points raised by Referee 4 and explain the modifications introduced in the revised version of the manuscript.

Referee: *It would be useful if the authors would extend the discussion of their results on a qualitative level because currently the paper is written in a too formal way. For example, it is not clear what are the physical consequences of the phase transition found by the authors. Namely, how does the avoided crossing between the third and the second excited states influence the microwave reflection coefficient, for example?*

The main link of our results to the ongoing experimental efforts is through the photonic spectral function (6), that shows the effect on the microwave spectroscopy produced by the avoided energy level crossings. In particular, the result in Fig.7 shows dispersions of the photon branch frequencies as a function of the flux biasing the SQUID analogous to the ones reported in the recent experiment [40] by Kuzmin et al. As shown in Fig. 6, without the interaction (i.e., $E_J \rightarrow 0$) with the junction the photonic branch energies increase monotonously with the ratio R_Q/Z . In contrast, with a finite E_J the dependence is not monotonous due to the anticrossing and the slope changes sign at the critical point $R_Q = Z$. A remarkable property of this transition is that there is a repercussion on all energy levels, not only the lowest excited ones.

Referee: *From the discussion of Fig. 7 in the text I understood that the phase transition implies the switching from the capacitive to the inductive response at low frequencies. It is a very clear and intuitive picture of the phase transition. However, in the conclusions the authors state that this issue is very subtle and sensitive to the circuit design etc. Can one say something definite strictly for the circuit shown in Fig. 1a?*

Referee 4 is right, the transition implies a shift from inductive to capacitive behaviour at finite frequency. Practically, this occurs at frequencies small enough with respect to the plasma frequency of the resonator where the mode frequencies are equally spaced. The change in the mode dispersion is well visible for finite-size systems. This definitely holds for

the circuit of Fig.1(a).

What we think is still a subtle point, dependent on the circuit design, is the characterization of the junction as insulating/superconducting in terms of its transport properties. To probe this, one approach would be to bias it with an external circuit providing a current drive. In principle, this could perturb the spectrum of the system and possibly also some key features giving rise to the transition.

Referee: *Does the junction in this circuit act as a capacitor in the insulating phase and as an inductor in the superconducting phase if the transmission line is sufficiently long?*

The sign of the shift for photonic branch dispersion can be interpreted in terms of an effective capacitive or inductive response. It is a property that emerges already for relative short transmission lines, as we have shown in our exact diagonalization results.